# Abnormal Intracranial Pulse Pressure Amplitude Despite Normalized Static Intracranial Pressure in Idiopathic Intracranial Hypertension Refractory to Conservative Medical Therapy

**DOI:** 10.3390/life11060537

**Published:** 2021-06-09

**Authors:** Per Kristian Eide

**Affiliations:** 1Institute of Clinical Medicine, Faculty of Medicine, University of Oslo, 0316 Oslo, Norway; p.k.eide@medisin.uio.no or peide@ous-hf.no; 2Department of Neurosurgery, Oslo University Hospital—Rikshospitalet, 0424 Oslo, Norway

**Keywords:** idiopathic intracranial hypertension, pseudotumor cerebri, benign intracranial hypertension, pulsatile intracranial pressure, cerebrospinal fluid, pulsation absorber mechanisms

## Abstract

Idiopathic intracranial hypertension (IIH) incorporates symptoms and signs of increased intracranial pressure (ICP) and is diagnosed by increased lumbar cerebrospinal fluid pressure. However, our knowledge about the characteristics of ICP abnormality, e.g., changes in pulsatile versus static ICP, remains scarce. This study questioned how overnight pulsatile ICP (mean ICP wave amplitude, MWA) associates with static ICP (mean ICP) in IIH patients who were refractory to conservative medical treatment. The material included 80 consecutive IIH patients undergoing ICP monitoring prior to shunt, as part of work-up for failed conservative medical therapy. In this group, the overnight mean ICP was normalized in 52/80 patients, but with abnormal overnight MWA in 45 of the 52 patients. Even though there was a positive correlation between MWA and mean ICP at group level and within individual ICP recordings, the levels of MWA were abnormal in a high proportion of patients despite normalized mean ICP. Taken together, the present results disclosed lasting abnormal pulsatile ICP despite normalized static ICP in IIH patients refractory to conservative medical therapy, which may reflect the underlying pathophysiology. It is tentatively suggested that abnormal pulsatile ICP in IIH may reflect alterations at the glia–neurovascular interface, resulting in impaired astrocytic pulsation absorber mechanisms.

## 1. Introduction

Idiopathic intracranial hypertension (IIH) is a brain disease characterized by symptoms and signs of increased intracranial pressure (ICP), including headache, visual failure, papilledema and sometimes sixth nerve palsy [1,2], but also symptoms such as olfactory dysfunction, pulsatile tinnitus and cognitive impairment [2,3,4,5,6]. The demonstration of elevated lumbar CSF pressure (>25 cm H_2_O) by lumbar puncture is a requirement for the diagnosis, though the composition of CSF should be normal [1,2]. The increased ICP in IIH may have several causes; recently, attention was given to the role of venous obstruction and increased cerebral venous pressure in IIH [1]. Current treatment strategies aim at reducing the increased ICP, either by conservative measures, e.g., weight reduction, or by medical-induced reduction in CSF production, e.g., acetazolamide medication [1,2,6]. However, some IIH patients are refractory to conservative medical treatment and become candidates for surgical interventions such as shunt surgery, optical nerve sheet fenestration or the stenting of dural sinus veins [1,7]. Despite the various treatment alternatives, a significant proportion of IIH patients have lasting symptoms, frequent hospital re-admissions, and a high proportion of patients become unable to maintain their occupancy [8,9]. A deeper understanding of the pathophysiology of IIH is needed to provide more efficient treatment.

Even though elevated ICP is the main hallmark of IIH, the existing knowledge about ICP abnormalities in IIH is scarce. The lumbar CSF pressure is diagnostic for IIH [1], but this measure may differ substantially from direct measurements of overnight ICP [10]. In addition, the lumbar CSF pressure refers to the static pressure component, while the ICP incorporates both a static and a pulsatile pressure component [11]. The latter refers to the pressure fluctuations occurring over the cardiac beat, relating to the pulsatile arterial blood pressure. How the pulsatile ICP associates with the static ICP in IIH needs to be better understood.

In our institution, IIH patients who are candidates for shunt surgery due to refractory response to conservative medical treatment undergo overnight ICP monitoring on a clinical routine basis [12]. We measure both the pulsatile and static ICP scores to establish information about the degree of intracranial hypertension, and the degree of abnormal pulsatile ICP (expressed as the mean ICP wave amplitude, MWA). This is of interest, not least because the MWA associates more strongly with the intracranial compliance (pressure–volume reserve capacity) than the static ICP (mean ICP) [13]. Moreover, in a recent study of IIH patients, we found that increased MWA was accompanied with evidence of impaired glymphatic function [14]. Compared with reference patients, IIH patients demonstrated stronger brainwide enrichment of a tracer administered to the CSF as well as delayed clearance of the tracer from a wide range of brain regions in IIH [14]. In vivo evidence of impaired glymphatic function was also found in patients with idiopathic normal pressure hydrocephalus (iNPH) [15], who like IIH patients demonstrate abnormal pulsatile ICP (MWA) despite normal static ICP (mean ICP) [16], indicative of impaired intracranial compliance despite normal static ICP. The glymphatic (or glia–lymphatic) system refers to a perivascular transport route for fluids and solutes driven by convective pulse pressure forces created by the arterial blood pressure pulsations [17,18]. According to this concept, CSF in the subarachnoid space is in direct communication with CSF along the blood vessels (perivascular) and the interstitial fluid of the brain, thus playing a crucial role in the clearance of soluble waste products from brain metabolism.

The recent observations of abnormal pulsatile ICP accompanying impaired glymphatic function may point to a more instrumental role of abnormal pulsatile ICP in IIH than previously acknowledged. Therefore, further characterization of changes in pulsatile ICP in IIH is warranted. The aim of the present study was to examine the association between pulsatile and static ICP in IIH, specifically asking to which extent the pulsatile ICP is abnormal despite normalized ICP in IIH patients refractory to conservative medical treatment.

## 2. Materials and Methods

### 2.1. Approvals and Study Design

The patient data and ICP scores were retrieved from a Neurovascular-Hydrocephalus quality registry approved by the Institutional Review Board of Oslo University Hospital (2011/6692). The ICP recordings included in the registry were obtained as part of routine clinical work-up prior to shunt surgery in IIH patients being refractory to conservative medical treatment.

### 2.2. IIH Patients

Patients included here fulfilled the diagnostic criteria of IIH, including increased lumbar opening pressure (>25 mmH_2_O), normal CSF composition, papilledema, normal neurological examination (except for 6th cranial nerve affection) and normal MRI with exclusion of venous thrombosis [1]. Due to failed conservative medical treatment, they were referred from local neurological and ophthalmological departments to the Department of Neurosurgery, Oslo University Hospital—Rikshospitalet, for shunt surgery. This department’s clinical routine is overnight ICP monitoring prior to shunt surgery to assess the severity of intracranial hypertension despite conservative medical therapy. For this reason, medical treatment for IIH was not ended before overnight ICP monitoring; patients were kept on their ordinary medication during ICP monitoring.

### 2.3. Measurements of Pulsatile and Static ICP

Overnight ICP monitoring, incorporating both the pulsatile and static ICP measures, was performed as previously described [12,19]. The ICP is measured via a solid pressure sensor (Codman ICP micro sensor placed in the parenchyma) via a small burr hole, with online measurements of the static ICP (mean ICP) and pulsatile ICP (mean ICP wave amplitude; MWA) from the same ICP recording [16]. Measurements do not involve CSF drainage or any placement of catheter to the CSF space. The MWA is computed over 6-s time windows, and refers to the pressure changes occurring during the cardiac cycle (Figure 1). During overnight ICP monitoring, average values of MWA and mean ICP were computed, as well as the percentage of 6-s time windows with MWA ≥ 5 mmHg or mean ICP ≥ 15 mmHg. In the present study, overnight average values were compared for the entire cohort of 80 IIH patients. In addition, the individual 6-s time windows were examined for eight of the patients.

All ICP recordings reported here were obtained prior to interventions such as shunt surgery. During ICP monitoring, patients were kept in their bed, lying flat during overnight monitoring. From our previous studies in patients with IIH, idiopathic normal pressure hydrocephalus (iNPH) or Chiari I malformation [12,16,19], we have categorized overnight pulsatile ICP scores as abnormal when the average MWA is ≥4 mmHg overnight and the percentage of MWA ≥ 5 mmHg exceeds 10% [12,16].

### 2.4. Statistical Analyses

Continuous data are presented as mean and standard deviation. The normal distribution of data was examined in both groups. Differences in continuous variables between the groups were determined by Pearson Chi-square test for categorical data and by independent samples t-test for continuous data. The statistical analysis was performed using SPSS version 27 (IBM Corporation, Armonk, NY, USA). Statistical significance was accepted at the 0.05 level (two-tailed).

## 3. Results

### 3.1. Patients

The study includes 80 consecutive patients who had fulfilled the diagnostic criteria of IIH disease [1], in whom overnight ICP measurements were performed prior to shunt surgery due to failed conservative medical treatment. Demographic information about the patient cohort is presented in Table 1.

### 3.2. Pulsatile Versus Static ICP at Group Level

The number of patients within the various MWA and mean ICP categories is presented in Table 2. Among the 52/80 (65%) patients with normalized static ICP (i.e., mean ICP < 15 mmHg), MWA was borderline (i.e., 4–5 mmHg) or abnormal (>5 mmHg) in 45/52 (87%) (Table 2).

The Pearson correlation coefficient between the overnight average of pulsatile ICP (MWA) and static ICP (mean ICP) was 0.51 (Figure 2a), and between percentages of MWA **≥** 5 mmHg and mean ICP **≥**15 mmHg was 0.45 (Figure 2b).

Figure 3a presents the overnight average MWA scores for different categories of mean ICP, demonstrating that the average MWA was above a threshold of 4 mmHg for all mean ICP categories. Moreover, the percentage of MWA ≥ 5 mmHg was above the threshold of 10% for all mean ICP categories (Figure 3b). Therefore, at the group level, the pulsatile ICP, expressed as the MWA, was elevated in IIH patients with failed conservative medical treatment despite normalization of mean ICP.

### 3.3. Pulsatile Versus Static ICP at the Individual Level

Since overnight average values for mean ICP and MWA may hide pressure variations, the distribution of MWA for various levels of mean ICP was determined for 6-s windows of individual ICP recordings in eight IIH patients. The MWA values for different mean ICP categories in the eight individual patient categories are shown in Figure 4. It is shown that the levels of MWA were elevated despite the normalization of mean ICP (mean ICP < 15 mmHg).

## 4. Discussion

The main observation of this study was lasting elevated pulsatile ICP (MWA) despite normalized static ICP (mean ICP) in IIH patients who were refractory to conservative medical therapy. The lasting abnormal pulsatile ICP may be of major importance in this patient group.

The present IIH patients fulfilled the criteria of IIH. The reason why the majority of patients had normalized static mean ICP at the time of ICP monitoring is that they had undergone conservative treatment, and our management strategy is not to end ongoing medication prior to ICP monitoring. The goal with overnight ICP monitoring is to obtain information about the degree of intracranial hypertension despite conservative measures to determine the need for shunt surgery. Accordingly, the present pulsatile/static ICP scores do not reflect the ICP at the time of diagnosing IIH, but the situation prior to surgical intervention.

The presently reported patients represent a subgroup of IIH as they were all referred to neurosurgery after failed conservative medical therapy. Provided an incidence of IIH of about 4–5 per 100,000 (or higher) [20,21], most IIH patients in our region are treated with conservative medical treatment. Our IIH patients are referred from local neurological and ophthalmological departments due to lasting symptoms and signs of intracranial hypertension following failed conservative medical treatment (typically weight reduction and medications such as acetazolamide). This department is the only neurosurgical service treating IIH patients for a population of about 3.1 million people. Annually, about 10–12 IIH patients are managed with first-time neurosurgery (i.e., shunt surgery). Therefore, the present group of IIH patients represents a selected cohort, and probably the most treatment-refractory IIH patients.

Exact normal pulsatile and static ICP scores have not been established, related to the fact that ICP monitoring requires a surgical procedure that cannot be performed in healthy individuals [11]. With regard to IIH, the ordinary procedure is to measure the lumbar CSF opening pressure with the patient in the lateral decubitus position (upper normal pressures ranging between 200 and 250 mm H_2_O, corresponding to 14.7–18.4 mm Hg) [22]. Today’s diagnostic criterion for IIH is a lumbar CSF pressure above 25 cm H_2_O [1]. On this basis, an overnight mean ICP < 15 mmHg may be considered as “normalized” mean ICP in IIH patients. With regard to the pulsatile ICP, we consider an average of overnight MWA < 4 mmHg to be within the normal range, while an average of overnight MWA 4 to 5 mmHg is borderline [12,16]. These figures are based on measuring pulsatile ICP in various patient cohorts, and assessing the pulsatile ICP scores most likely being accompanied with shunt response. In this cohort, 52/80 patients presented with an average of overnight mean ICP < 15 mmHg, in whom the overnight MWA was ≥4 mmHg in 45/52 (87%) patients.

The present observations showed a positive correlation between MWA and ICP, though the MWA could not be described solely by the mean ICP since correlation coefficients were 0.51 or lower. It may therefore not be argued that the ICP wave amplitude is derived directly from the mean ICP (which would imply that low mean ICP would correspond with low MWA and vice versa). On the contrary, the present results document abnormal pulsatile ICP (MWA) despite normalized mean ICP in a substantial proportion of IIH patients. Elevated MWA was seen for both low and high mean ICP levels. The present data add evidence to previous studies from our group of abnormal pulsatile ICP in IIH patients [12,19].

What might be the reasons for abnormal pulsatile ICP (MWA) despite normal static ICP (mean ICP)? Impaired intracranial compliance, i.e., reduced pressure–volume reserve capacity, may be one underlying cause. It has previously been shown that elevated pulsatile ICP (MWA) may reflect impaired intracranial compliance better than static ICP (mean ICP) [13]. The intracranial compliance may be impaired despite a normal mean ICP [23]. On the other hand, the biological mechanisms underlying impaired intracranial compliance are less understood.

With regard to a biological basis for abnormal pulsatile ICP in IIH, the glia–neuro-vascular interface, also denoted neuro-vascular unit or neuro-vascular coupling, may be of particular significance. It has been estimated that an adult human brain comprises over 100 billion capillaries, with an estimated total perfused length of about 600–700 km and a total surface area of 20 m^2^ [24,25]. The astrocytic endfeet surround the basal lamina (or basement membrane) and the endothelial cells and pericytes of the capillaries, the basal lamina comprising a loose matrix wherein fluids and solutes may be transported along the blood vessels (perivascular). The astrocytic endfeet ensheath the cerebral capillaries, as previously shown by three-dimensional (3D) electron microscopy (EM) [26], forming a continuous donut-shaped structure around the capillaries, only separated by inter-endfeet gaps of about 20 nm. Most importantly, the fluid content and volume of the astrocytic endfeet are regulated by fast efflux and influx of water along the water channel aquaporin-4 (AQP4), which constitutes about 50% of the area of endfeet facing the basal lamina [27]. The physiological role of the astrocytic endfeet is debated, but it may be proposed that the volume regulation of the endfeet may act as a pulsation absorber for the arterial blood pressure pulsations created by cardiac contractions. In previous studies, we proposed the existence of a cardiac pulsation absorbance from a biophysical and pressure signaling perspective [28], but without an underlying biological perspective.

Several lines of evidence indicate that the function and volume regulation of astrocytic endfeet may be affected in IIH. From cortical brain biopsies of IIH patients, we demonstrated pathological alterations in the basal lamina and pericytes of the cerebral capillaries [29], the loss of BBB integrity and the leakage of the pro-inflammatory fibrinogen from blood to the perivascular space [30], and patchy astrogliosis [31]. Moreover, transmission electron microscopy showed pathological mitochondria in astrocytic perivascular endfeet as well as shortened endoplasmic-reticular-contact sites, suggestive of cellular metabolic failure [32]. An increased astrocytic perivascular aquaporin-4 (AQP4) expression in light microscopy indicated a higher demand on fluid exchange, causing a compensatory increase in perivascular AQP4 [31], though not by immunogold electron microscopy [30]. Taken together, the observed alterations in astrocytic endfeet of IIH may be an anatomical basis for impaired astrocytic pulsation absorber capacity in IIH. It may as well represent an anatomical basis for the impaired glymphatic function recently reported in IIH [14]. Future research is needed to further clarify these mechanisms.

Some limitations of this study should be noted. Normal values of mean ICP and MWA are not available as ICP monitoring is invasive and cannot be performed in healthy volunteers. Moreover, since this study is a case series of IIH patients referred to neurosurgery for conservative medical treatment-resistant IIH, caution should be made when extrapolating results to IIH patients in general. Finally, it cannot be concluded from the present data what causes increased ICP in IIH, for example, the role of increased intracranial venous pressure.

## 5. Conclusions

This study showed that a significant proportion of IIH patients being refractory to conservative medical therapy presented with abnormal pulsatile ICP (increased MWA) despite normalized static ICP (mean ICP). The abnormal pulsatile ICP may be one characteristic of IIH disease that is refractory to conservative medical treatment. It is tentatively proposed that abnormal pulsatile ICP reflects the underlying IIH pathophysiology, possibly resulting from alterations and the glia–neurovascular interface that impair the astrocytic pulsation absorber mechanisms. This hypothesis should be explored in future studies.

## Figures and Tables

**Figure 1 life-11-00537-f001:**
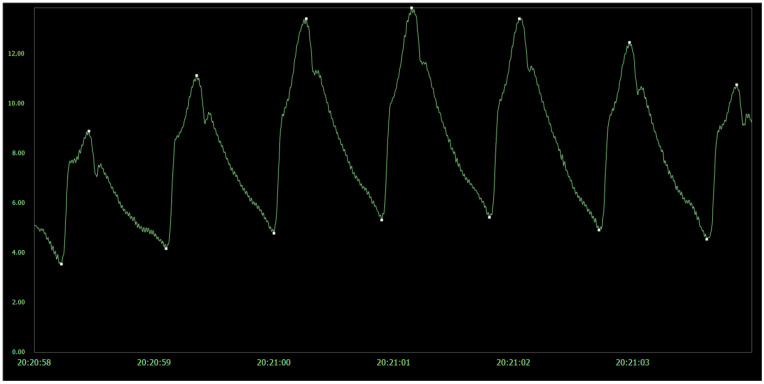
The pulsatile and static ICP is derived from the same ICP signal, here presented with ICP on y scale (mmHg) and time on x axis. The pulsatile ICP, referred to as Mean Wave Amplitude (MWA), describes pressure changes occurring during the cardiac cycle, i.e., increase in pressure from diastolic minimum pressure to systolic maximum pressure. The static ICP (mean ICP), on the other hand, is the absolute pressure value measured against a reference pressure, not considering the pressure changes occurring during the cardiac cycle. The MWA and the mean ICP were determined over consecutive 6-s time windows; here, mean ICP was 8.2 mmHg and MWA 7.4 mmHg.

**Figure 2 life-11-00537-f002:**
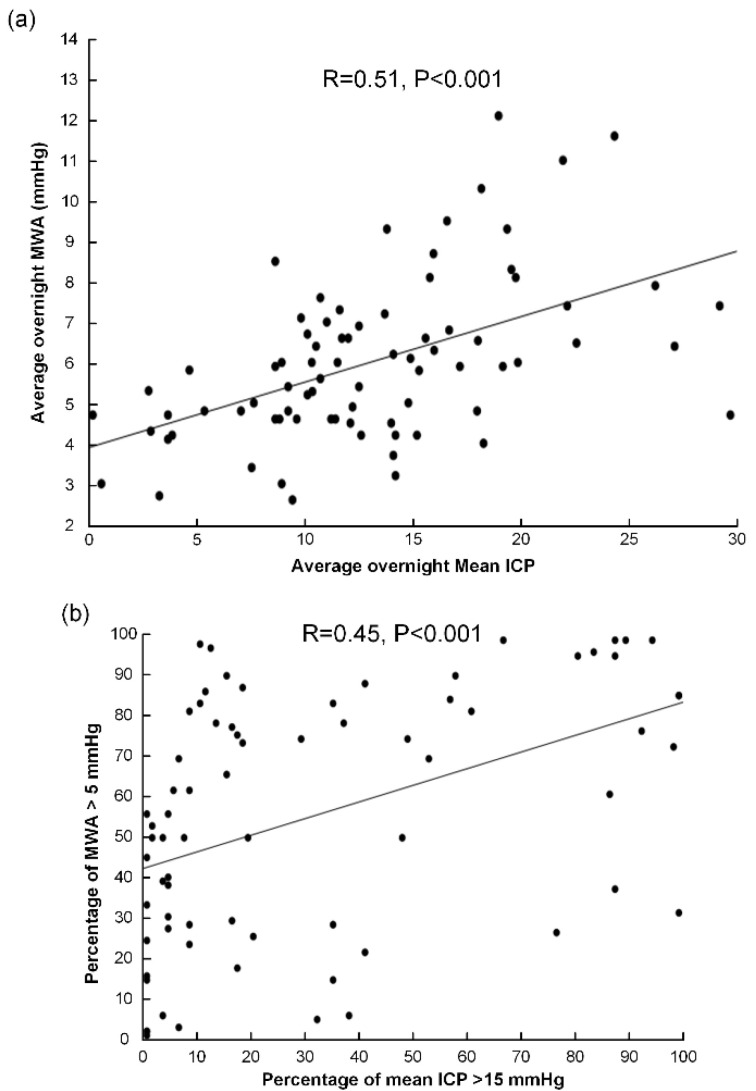
In the cohort of 80 IIH patients, the Pearson correlation was determined between (**a**) the average values of overnight scores of mean ICP and MWA and (**b**) the percentage occurrence of MWA ≥ 5mmHg versus percentage occurrence of mean ICP ≥ 15 mmHg. Each plot shows the fit line and Pearson correlation coefficient. The ICP was measured prior to any intervention in IIH patients who were refractory to conservative medical therapy.

**Figure 3 life-11-00537-f003:**
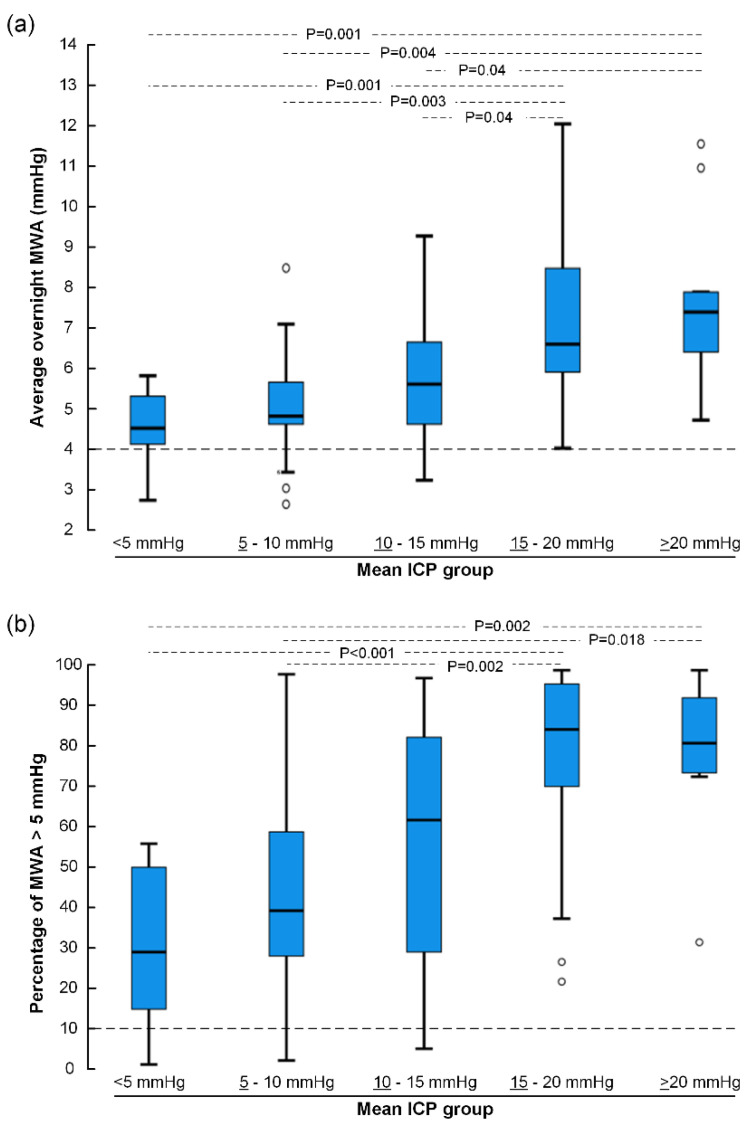
The overnight ICP recordings of the cohort of 80 IIH patients were categorized according to mean ICP level; for each mean ICP category (**a**) the overnight MWA was determined as well as (**b**) the overnight percentage of MWA ≥ 5 mmHg. Differences between groups were determined by one-way ANOVA with Bonferroni-corrected post hoc tests. Horizontal lines indicate that overnight MWA ≥ 4 mmHg and 10% of MWA ≥ 5 mmHg are considered as abnormal.

**Figure 4 life-11-00537-f004:**
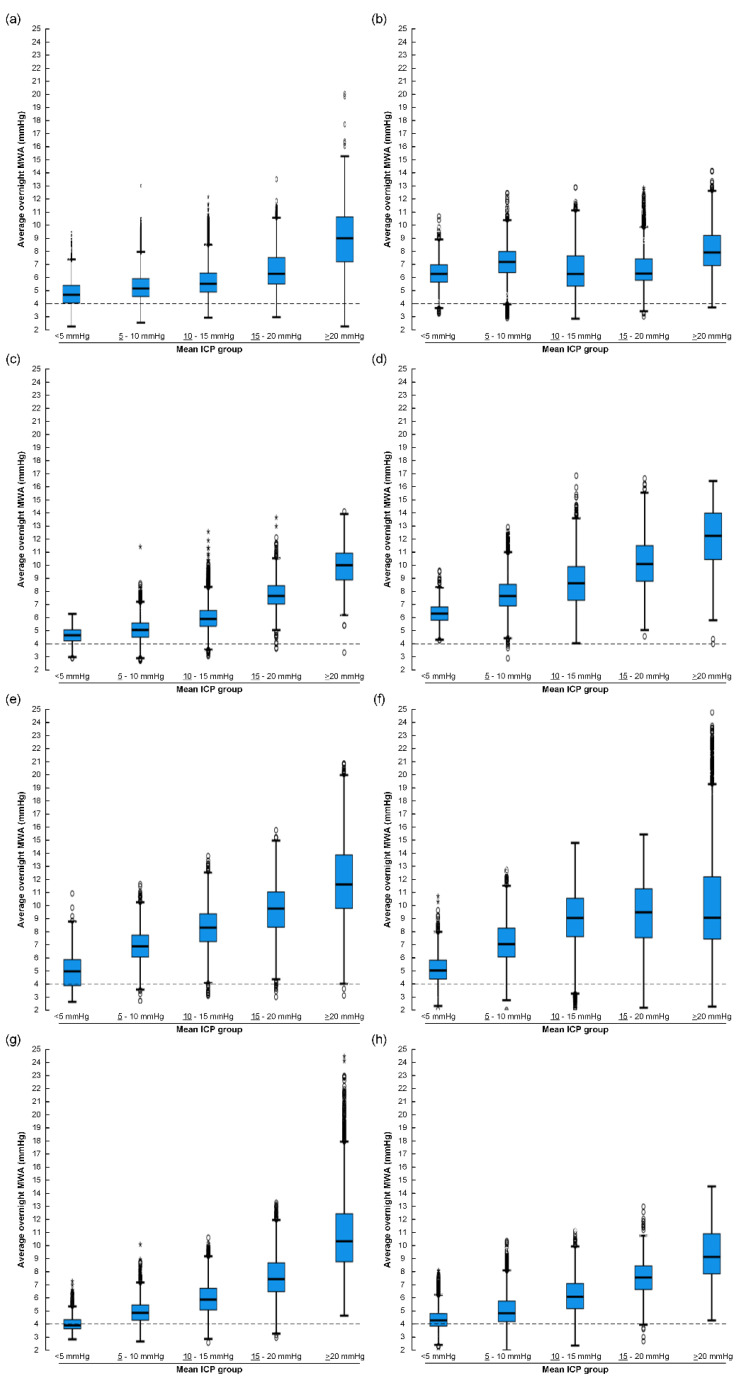
For eight of the 80 IIH patients, the ICP recordings were categorized according to mean ICP and MWA levels for the individual 6-s time windows of each patient recording. For each mean ICP category, the overnight MWA was determined: (**a**) PatID 1 (43,282 6-s time windows). (**b**) PatID 2 (27,962 6-s time windows). (**c**) PatID 3 (26,800 6-s time windows). (**d**) PatID 4 (24,795 6-s time windows). (**e**) PatID 5 (21,524 6-s time windows). (**f**) PatID 6 (25,369 6-s time windows). (**g**) PatID 7 (27,360 6-s time windows). (**h**) PatID 8 (25,946 6-s time windows).

**Table 1 life-11-00537-t001:** Information about the IIH patients.

Variable	IIH Patients
**N**	80
***F (N, %)***	67 (84%)
***M (N, %)***	13 (16%)
***Age (years)***	30.6 ± 13.5
***BMI (kg/m^2^)***	30.1 ± 7.0

F: Female; M: Male; BMI: Body mass index.

**Table 2 life-11-00537-t002:** Number of patients with overnight MWA and mean ICP scores within different categories.

MWA Category (mmHg)	Mean ICP Category (mmHg)					
	**<5**	**≥5–<10**	**≥10–<15**	**≥15–<20**	**≥20**	**Total (N, %)**
**<4**	2	3	2	-	-	7 (8.8%)
**≥4–<5**	5	6	7	3	1	22 (27.5%)
**≥5**	3	6	18	16	8	51 (63.8%)
**Total (N, %)**	10 (12.5%)	15 (18.8%)	27 (33.8%)	19 (23.8%)	9 (11.3%)	80 (100%)

ICP: Intracranial pressure; MWA: Mean ICP wave amplitude.

## Data Availability

The data presented in this work are available upon reasonable request.

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
