# Peer review of "Abnormal Intracranial Pulse Pressure Amplitude Despite Normalized Static Intracranial Pressure in Idiopathic Intracranial Hypertension Refractory to Conservative Medical Therapy"

_life, 2021, doi:10.3390/life11060537_

Round 1

Reviewer 1 Report

I reccommend acceptance

Author Response

Thanks to the reviewer for thorough assessment of this work.

Reviewer 2 Report

First of all, congratulations on your work! The manuscript is very well documented, scientifically sound, and presents an interesting topic of research. The paper documents that lasting elevated pulsatile ICP despite normalized static ICP may be a significant factor to take into account in IIH patients who are refractory to conservative medical therapy.

I have one question: from your experience are there any particular clinical features encountered in patients with lasting elevated pulsatile, to help us in early identifying the subgroup of patients with IIH that most probably will need surgery?

Author Response

I greatly appreciate the reviewer's assessment and thoughtful comments. Unfortunately, so far I have not identified any clinical features indicative of abnormal pulsatile ICP. We are assessing possible biomarkers from optical coherence tomography (OCT), but it would be premature to comment on this in the present paper. This aspect is something we work on, as it is highly needed with non-invasive measures of abnormal pulsatile ICP.

This manuscript is a resubmission of an earlier submission. The following is a list of the peer review reports and author responses from that submission.

Round 1

Reviewer 1 Report

The authors present a retrospective series of 80 patients with medically refractory IIH who underwent overnight ICP monitoring prior to shunting. 

There are a number of limitations to this manuscript.

First, IIH is now known to be due to elevated intracranial venous pressures. There is a direct correlation between venous sinus pressures and ICP on spinal tap in IIH patients. There is essentially no mention of this relationship throughout the entirety of the manuscript, almost as if the authors are not aware of these revelations. There is a huge literature base in the neurosurgical and neurointerventional specialties that have studied intracranial venous hypertension as it relates to transverse sinus stenosis and cerebral venous outflow impairment as well as elevated central venous pressures from obesity or other co-morbid conditions. 

Second, intracranial venous pressures, which are known to be the driver of elevated CSF pressures, are affected by physiologic parameters, most notably BP and end-tidal CO2. The authors do not even attempt to correlate SBP/DBP/MAP to ICP changes nor do they identify periods of sleep and its effect on pressures. ICP are known to spike in patients with obstructive sleep apnea while sleeping but the authors do not account for waking or asleep measurements, even though they had overnight monitoring. There is essentially no statistical control for other important confounders like gender, BMI, medication dose, MRI/MRV findings like venous sinus dominance, presence/absence of transverse sinus or jugular stenosis.

Third, it is unclear how the pulsatility results that are presented are useful at all to physicians managing this condition. Conjectures are made regarding why (or how) this might be the case but these observations have almost no clinical utility. 

Fourth, as a neurosurgeon that regularly performs stenting and shunting for this condition, I cannot support subjecting patients to the additional risk, cost, and LOS for ICP monitoring prior to shunting. In modern clinical practice shunting is viewed as a "last resort" treatment for patients with severe symptoms that fail stenting or are not candidates for venous sinus stenting. Obviously there are differences in management strategies at different centers in different countries, but ICP monitoring, in general, has a limited role in most patients with IIH. Subjecting all patients to this procedure unnecessarily should not be supported.

Author Response

Comments to the report of “Reviewer 1” (life-1148298)

General Comments #1

The authors present a retrospective series of 80 patients with medically refractory IIH who underwent overnight ICP monitoring prior to shunting. 

There are a number of limitations to this manuscript.

Response to general comments #1

The reviewer’s comments are commented on below.

Specific comment #1

First, IIH is now known to be due to elevated intracranial venous pressures. There is a direct correlation between venous sinus pressures and ICP on spinal tap in IIH patients. There is essentially no mention of this relationship throughout the entirety of the manuscript, almost as if the authors are not aware of these revelations. There is a huge literature base in the neurosurgical and neurointerventional specialties that have studied intracranial venous hypertension as it relates to transverse sinus stenosis and cerebral venous outflow impairment as well as elevated central venous pressures from obesity or other co-morbid conditions.

Response to specific comment #1

While the relationship between venous and intracranial pressures are of great interest and worth studying, the present study did not address these pressures. It was beyond the scope of the work. In this work, ICP scores measured overnight were highlighted. It is not feasible to measure intracranial venous pressures over lon time spans. The simultaneous CSF and venous pressures are measured during interventional procedures, which is a complete different setting. The statement by the reviewer that “IIH is now known to be due to elevated intracranial venous pressures” is up to debate. An alternative view is that increased ICP in IIH is caused by biological changes at the glia-vascular interface, as detailed at the end of the discussion (page 8, line 201 to page 9, line 213).

Specific comment #2

Second, intracranial venous pressures, which are known to be the driver of elevated CSF pressures, are affected by physiologic parameters, most notably BP and end-tidal CO2. The authors do not even attempt to correlate SBP/DBP/MAP to ICP changes nor do they identify periods of sleep and its effect on pressures. ICP are known to spike in patients with obstructive sleep apnea while sleeping but the authors do not account for waking or asleep measurements, even though they had overnight monitoring. There is essentially no statistical control for other important confounders like gender, BMI, medication dose, MRI/MRV findings like venous sinus dominance, presence/absence of transverse sinus or jugular stenosis.

Response to specific comment #2

The reviewer address various aspects about CSF and intracranial pressures that were not addressed in this work The aim of the present work was to compare the pulsatile (i.e. MWA) and static (mean ICP) ICP scores; both these scores are retrieved from the same ICP recording and from the same patient. In that sense, each patient was its own control. Why is this relevant for IIH? Because the pulsatile ICP (MWA) reflects better the intracranial compliance (pressure-volume reserve capacity) than the static ICP (mean ICP). The lesson from the present results is that intracranial compliance may be impaired despite normalized mean ICP. The reason for abnormal intracranial compliance needs to be studied further. The statement that intracranial venous pressure is the driver of elevated CSF pressure is presently a hypothesis and not an established fact. As commented above, I would argue for other causes for increased ICP scores than elevated venous pressure. Since IIH pathophysiology is largely unknown, it is very important to address several and different mechanisms.

Specific comment #3

Third, it is unclear how the pulsatility results that are presented are useful at all to physicians managing this condition. Conjectures are made regarding why (or how) this might be the case but these observations have almost no clinical utility. 

Response to specific comment #3

In our institution, we have used ICP monitoring as described here with assessment of pulsatile /static ICP on a clinical routine basis for more than 15 years. Dedicated software is used that enables the physicians to get immediate information about pulsatile ICP scores, which is used in patient management. Obviously, this clinical practice would not be continued if there was “no clinical utility”.

Specific comment #4

Fourth, as a neurosurgeon that regularly performs stenting and shunting for this condition, I cannot support subjecting patients to the additional risk, cost, and LOS for ICP monitoring prior to shunting. In modern clinical practice shunting is viewed as a "last resort" treatment for patients with severe symptoms that fail stenting or are not candidates for venous sinus stenting. Obviously there are differences in management strategies at different centers in different countries, but ICP monitoring, in general, has a limited role in most patients with IIH. Subjecting all patients to this procedure unnecessarily should not be supported.

Response to specific comment #4

As commented on in the Methods section (page 2, line 77), ICP monitoring was done in patients with failed conservative-medical treatment, and is not advocated in “all” IIH patients. We have, however, found the use of ICP monitoring as described here highly useful in this patient group. The present paper is no presentation of cons / pros regarding stenting / shunting / ICP monitoring, but presents data this author considers important for understanding IIH pathophysiology.

Reviewer 2 Report

Transverse sinus stenosis is accepted as the most sensitive radiological hallmark of the diagnosis of Idiopathic intracranial hypertension (IIH) , having a sensitivity of 84.4%. Do you have the data concerning this radiological feature for the present cohort of patients? If data are available, i recommend the authors to add these data.

Author Response

Comments to the report of Reviewer 2 (life-1148298)

Specific comment #1

Transverse sinus stenosis is accepted as the most sensitive radiological hallmark of the diagnosis of Idiopathic intracranial hypertension (IIH) , having a sensitivity of 84.4%. Do you have the data concerning this radiological feature for the present cohort of patients? If data are available, i recommend the authors to add these data.

Response to specific comment #1

Identification of imaging biomarkers of IIH is of definite interest and importance, but was not the focus of this work. We do not have radiological data about sinus vein stenosis in the present IIH patients, and cannot answer these questions.

Reviewer 3 Report

Professor Eide analyzed 80 consecutive symptomatic patients affected by Idiopathic Intracranial Hypertension (IIH) non-responsive to conservative-medical therapy, already selected by neurologists and ophthalmologists for shunt surgery. All patients fulfilled diagnostic criteria of IIH.

All patients continued drug therapy and was submitted to overnight ICP monitoring prior to surgery to assess severity of intracranial hypertension by means of an intraparenchymal micro sensor. 52/80 patients had normal ICP, probably thanks to medical therapy, but 45/52 had abnormal mean ICP wave amplitude, suggesting an abnormal pulsatile ICP despite the normalized mean ICP. The majority of patients showed ICP < 20 mmHg, and only 8 patients had ICP >20 mmHg; 91.2% of all patients showed an abnormal pulsatile ICP, but in 8.8% of the patients the ICP and MWA was normal.

In my opinion, the results of this very interesting study suggest two questions.

  1. As reported by Mahdavi (Mahdavi ZK, et al. Association patterns of simultaneous intraventricular and intraparenchymal intracranial pressure measurements. Neurosurgery, 79(4), 561-67, 2016), intraparenchymal and intraventricular ICP values may not correlate, especially when ICP is < 25 mmHg, therefore ICP measurement detected by means of intraparenchymal sensor might not the same of the ventricular drainage. Why was the ICP and MWA measurement carry out only with intraparenchymal device?
  2. Did the relief of normal ICP and MWA in 8 patients modified surgical indications and/or strategies?

Author Response

Comments to the report of Reviewer 3 (life-1148298)

General Comment #1

Professor Eide analyzed 80 consecutive symptomatic patients affected by Idiopathic Intracranial Hypertension (IIH) non-responsive to conservative-medical therapy, already selected by neurologists and ophthalmologists for shunt surgery. All patients fulfilled diagnostic criteria of IIH.

All patients continued drug therapy and was submitted to overnight ICP monitoring prior to surgery to assess severity of intracranial hypertension by means of an intraparenchymal micro sensor. 52/80 patients had normal ICP, probably thanks to medical therapy, but 45/52 had abnormal mean ICP wave amplitude, suggesting an abnormal pulsatile ICP despite the normalized mean ICP. The majority of patients showed ICP < 20 mmHg, and only 8 patients had ICP >20 mmHg; 91.2% of all patients showed an abnormal pulsatile ICP, but in 8.8% of the patients the ICP and MWA was normal.

In my opinion, the results of this very interesting study suggest two questions.

Response to general comment #1

This is a precise overview of the work. The reviewers interest is greatly appreciated.

Specific comment #1

  1. As reported by Mahdavi (Mahdavi ZK, et al. Association patterns of simultaneous intraventricular and intraparenchymal intracranial pressure measurements. Neurosurgery, 79(4), 561-67, 2016), intraparenchymal and intraventricular ICP values may not correlate, especially when ICP is < 25 mmHg, therefore ICP measurement detected by means of intraparenchymal sensor might not the same of the ventricular drainage. Why was the ICP and MWA measurement carry out only with intraparenchymal device?

Response to specific comment #1

In acute situations, there may be gradients in pulsatile and static ICP, which are accompanied with displacement of brain tissue, as recently commented on in a review about ICP monitoring (Evensen & Eide, FBCNS, 2020). In a chronic condition such as IIH, gradients in pressure would not be expected as it would cause brain tissue displacement that would be life-threatening. Our routine use of parenchymal ICP sensor (page 3, line 85) is also based on previous studies that have demonstrated identical MWA scores, whether measured within the ventricular CSF or the parenchyma (e.g. see Eide & Sæhle, Acta Neurochir, 2010). Even though the reviewer raises an interesting question, it would be beyond the scope of this present work to discuss measurement differences depending on location. Based on our previous work, we have no reason to believe that pulsatile ICP would differ whether measured in parenchyma or ventricles.

Specific comment #2

  1. Did the relief of normal ICP and MWA in 8 patients modified surgical indications and/or strategies?

Response to specific comment #2

Yes, in this institution, overnight monitoring of mean ICP/MWA is done on clinical routine basis, aiding patient management. It has been commented in the Introduction that this is done on clinical routine basis, meaning that ICP scores aid in patient management.

Round 2

Reviewer 1 Report

None. Authors argued with reviewer points but did not make any significant changes to the manuscript to address any of these comments. 

Author Response

No further comments